# ADVERSARIAL ATTACKS AS NEAR-ZERO EIGENVALUES IN THE EMPIRICAL KERNEL OF NEURAL NETWORKS

## ABSTRACT

Adversarial examples —imperceptibly modified data inputs designed to mislead machine learning models— have raised concerns about the robustness of modern neural architectures in safety-critical applications. In this paper, we propose a unified mathematical framework for understanding adversarial examples in neural networks, corroborating Szegedy et al. (2014)'s original conjecture that such examples are exceedingly rare, despite their presence in the proximity of nearly every test case. By exploiting Mercer's decomposition theorem, we characterise adversarial examples as those producing near-zero Mercer's eigenvalues in the empirical kernel associated to a trained neural network. Consequently, the generation of adversarial attacks, using any known technique, can be conceptualised as a progression towards the eigenvalue space's zero point within the empirical kernel. We rigorously prove this characterisation for trained neural networks that achieve interpolation and under mild assumptions on the architecture, thus providing a mathematical explanation for the apparent contradiction of neural networks excelling at generalisation while remaining vulnerable to adversarial attacks. We have empirically verified that adversarial examples generated for both fully-connected and convolutional architectures through the widely-known DeepFool algorithm and through the more recent Fast Adaptive Boundary (FAB) method consistently lead to a shift in the distribution of Mercer's eigenvalues toward zero. These results are in strong agreement with predictions of our theory.

## 1 INTRODUCTION

Adversarial examples are specially crafted input data points designed to cause a model to output an incorrect prediction. These examples are created by making small, imperceptible perturbations to the input data (e.g., images, text or audio) which are typically indistinguishable to humans but can have a significant impact on the model's output (Szegedy et al., 2014; Goodfellow et al., 2015). The existence of adversarial examples has raised questions about the robustness and reliability of deep learning models when used in safety-critical applications (Ruan et al., 2021) since empirical evidence shows that neural networks are particularly sensitive (Moosavi-Dezfooli et al., 2017).

Adversarial examples can occur in many domains, and reveal serious vulnerabilities. For example, a self-driving car's neural object detection system might misclassify a stop sign as a yield sign if an adversarial sticker is placed on it (Akhtar et al., 2021). In NLP, attackers can make subtle changes to text to deceive sentiment analysis or spam detection models (Zhang et al., 2020b) and, in cybersecurity, attackers can modify malware to evade detection by antivirus or intrusion detection systems (Rosenberg et al., 2021). As a result, researchers and practitioners have developed techniques to defend against adversarial attacks, such as adversarial training, input preprocessing, and robust model architectures (Zhang et al., 2020a; Fowl et al., 2021; Carlini et al., 2019; Han et al., 2023).

The foundational paper introducing adversarial examples (Szegedy et al., 2014) characterised them as "intriguing properties" of neural networks and raised a compelling question: how can neural networks exhibit strong generalisation performance on test examples drawn from a data distribution, while simultaneously being susceptible to adversarial examples? An interesting hypothesis, which we recapitulate next, was already outlined in the paper's concluding remarks.

*"A possible explanation is that the set of adversarial negatives is of extremely low probability, and thus is never (or rarely) observed in the test set, yet it is dense (much like the rational numbers), and so it is found near virtually every test case."*

Despite the substantial attention dedicated by researchers to explaining the prevalence of adversarial examples, as discussed in Section 6, and despite progress in identifying new attack variants and developing defensive approaches, robustness measures (Yu et al., 2019; Carlini et al., 2019; Wang et al., 2023), and theoretical guarantees based on such measures (Bhagoji et al., 2019; Shafahi et al., 2019), the fundamental nature of adversarial examples continues to elude complete understanding (Guo et al., 2018; Ilyas et al., 2019; Madry et al., 2018; Qin et al., 2019).

**Our Contribution**  In this paper, we provide novel theoretical results that further substantiate the original hypothesis by Szegedy et al. (2014). Our research relies on the formulation of neural networks as specific instances of parameter-dependent kernel machines, which has enabled researchers to leverage well-established results in the field of Kernel Theory (Györfi et al., 2002; Saitoh & Sawano, 2016) to investigate the generalisation properties of neural models (Berthier et al., 2020; Canatar et al., 2021; Simon et al., 2023). Specifically, a trained neural network can be equivalently viewed as realising a non-linear transformation of the input data, which is associated to an empirical kernel, followed by a linear transformation characterised by the readout weights.

Mercer's decomposition theorem (Mercer, 1909; Minh et al., 2006) is a fundamental result in Kernel Theory which plays a crucial role in various ML algorithms. Mercer's theorem states that, for a kernel function applicable to pairs of data points sampled from a probability distribution with compact support, the kernel's evaluation can be represented as an infinite sum of products. These products consist of the application of Mercer's eigenfunctions mapping the input data points to real values, weighted by Mercer's eigenvalues—positive scalars determining the relative contribution of each term in the decomposition. Such eigenvalues and eigenfunctions are specific to the kernel function and input data distribution, but remain independent from the specific data points being evaluated.

We adopt the perspective that the introduction of an adversarial example can be regarded as a modification of the empirical data distribution derived from the training set, expanded to incorporate the adversarial example. By Mercer's theorem, even when maintaining the same empirical kernel associated to a trained model, this process results in a new set of Mercer eigenvalues and eigenfunctions, since the data distribution used to compute the Mercer's decomposition is changed. Our first technical contribution is to characterise adversarial examples as those producing near-zero Mercer eigenvalues in the updated decomposition of the neural network's empirical kernel. This characterisation then allows us to show that adversarial examples have measure zero in the limit where they become infinitesimally close to any test example sampled from the original data distribution. Our results provide a rigorous mathematical explanation for what appears to be a paradoxical empirical observation: while neural networks demonstrate strong generalisation to novel test examples, they are also susceptible to adversarial attacks. Indeed, it follows from our results that adversarial examples are exceedingly unlikely to occur in the test set, thereby providing compelling support for the hypothesis by Szegedy et al. (2014). Our theory could also open the way to new approaches in designing defense mechanisms and detection methods for adversarial attacks (Tramer, 2022).

We conducted experiments on a fully-connected neural network trained on a subset of MNIST, and on a pretrained convolutional neural network on a subset of CIFAR10, both achieving interpolation on the training set and perfect generalisation performance on a subset of the test sets. We then used the DeepFool algorithm (Moosavi-Dezfooli et al., 2016) (respectively, the FAB attack (Croce & Hein, 2020)) on the FCN (respectively, the CNN) to generate adversarial examples, and used the numerical method by Baker (1977); Rasmussen & Williams (2006) to compute Mercer's decomposition on the empirical kernel associated with both the original empirical data distribution and our updated empirical data distributions, which incorporate the adversarial examples. We have analysed the relevant distributions of eigenvalues, computed those with minimal value and estimated the integral of relevant quantities near zero. Our numerical experiments align with our theory, demonstrating that adversarial examples induce a shift in the distribution of Mercer eigenvalues towards zero, and corroborate that our theory is indeed architecture- and data-agnostic.

## 2 PRELIMINARIES

We use standard notation for real-valued vectors $\mathbf{v} \in \mathbb{R}^n$, matrices $\mathbf{A} \in \mathbb{R}^{m \times n}$, and their transposes $\mathbf{v}^T$ and $\mathbf{A}^T$. As usual, $A_{i,j}$ denotes the $(i, j)$-element of $\mathbf{A}$ and $v_i$ the $i$-th element of $\mathbf{v}$. We denote as $\mathbf{a}_i$ the vector in the $i$-th row of $\mathbf{A}$ and $\mathbf{0}_n$ denotes the $n$-dimensional zero vector. For $\mathbf{A} \in \mathbb{R}^{m \times n}$ and $\mathbf{v} \in \mathbb{R}^n$, we denote with $\mathbf{A} : \mathbf{v}$ the matrix obtained by extending $\mathbf{A}$ with $\mathbf{v}$ as an additional row. The Moore-Penrose pseudo-inverse of $\mathbf{A}$ is denoted as $\mathbf{A}^\dagger$ and the trace of $\mathbf{A}$ as $\mathrm{Tr}(\mathbf{A})$.

**Neural networks.** A neural network with $L$ layers is a tuple $\mathcal{N} = \langle \{\Theta^\ell\}_{1 \leq \ell \leq L}, \{\sigma^\ell\}_{1 \leq \ell \leq L} \rangle$. Each layer $\ell \in \{1, ..., L\}$ consists of parameters $\Theta^\ell$ (matrices and vectors), and a non-linear operator $\sigma^\ell$. On input $\mathbf{x} \in \mathbb{R}^{N_0}$, $\mathcal{N}$ sets $\mathbf{x}^0 := \mathbf{x}$ and then computes, for each layer $1 \leq \ell \leq L$, a sequence of vector representations $\mathbf{x}^\ell = \sigma^\ell(\Theta^\ell, \mathbf{x}^{\ell-1})$. The output $\mathcal{N}(\mathbf{x})$ is given by $\mathbf{x}^L$. This description encompasses various common architectures including fully-connected networks (FCNs), convolutional neural networks (CNNs) and their variants. For instance, for FCNs, $\Theta^\ell$ consists of a weight matrix $\mathbf{W}^\ell \in \mathbb{R}^{N_\ell \times N_{\ell-1}}$ where $N_\ell$ is the width of the $\ell$-th layer and a bias $\mathbf{b}^\ell \in \mathbb{R}^{N_\ell}$, and $\sigma^\ell$ is an activation function, a real-valued function applied entrywise to a pre-activation vector $\mathbf{h}^\ell = \mathbf{W}^\ell \mathbf{x}^{\ell-1} + \mathbf{b}^\ell$. We denote as $N$ the size of $\mathbf{x}^{L-1}$. For the last layer, we assume $\Theta^L$ is a column matrix $\mathbf{W}^L \in \mathbb{R}^N$ and $\sigma^L(\Theta^L, \mathbf{x}^{L-1}) = (\mathbf{W}^L)^T \mathbf{x}^{L-1}$ (ensuring linearity and a real-valued output). In this setting, the weights $\mathbf{W}^L$ are referred to as the readout weights.

**Kernels.** A kernel on $\mathbb{R}^{N_0}$ is a positive semi-definite symmetric function $K : \mathbb{R}^{N_0} \times \mathbb{R}^{N_0} \mapsto \mathbb{R}$. By Mercer's theorem, given a distribution $p$ with compact support on $\mathbb{R}^{N_0}$, there exist unique countable collections of Mercer's eigenvalues $\lambda_i^{K,p} \geq 0$ and Mercer's eigenfunctions $\varphi_i^{K,p} : \mathbb{R}^{N_0} \mapsto \mathbb{R}$, for $i \in \mathbb{N}$, such that: $K(\mathbf{x}, \mathbf{x}') = \sum_i^\infty \lambda_i^{K,p} \varphi_i^{K,p}(\mathbf{x}) \varphi_i^{K,p}(\mathbf{x}')$ for all $\mathbf{x}, \mathbf{x}' \in \mathbb{R}^{N_0}$; and for $i, j \in \mathbb{N}$, we have $\mathbb{E}_{\mathbf{x} \sim p(\mathbf{x})} \left( \varphi_i^{K,p}(\mathbf{x}) \varphi_j^{K,p}(\mathbf{x}) \right) = \delta_{i,j}$, with $\delta_{i,j}$ the Kronecker Delta. The first condition represents the application of the kernel function to $\mathbf{x}$ and $\mathbf{x}'$ as an infinite sum of products, where the $i$-th product consists of the application of the $i$-th Mercer eigenfunction to $\mathbf{x}$ and $\mathbf{x}'$, mapping these data points onto real values, weighted by the $i$-th Mercer (nonnegative) eigenvalue. The second condition requires orthonormality of the Mercer eigenfunctions with respect to the data distribution. The density $\rho^{K,p}(\lambda)$ of Mercer's eigenvalues is a measure with support on $\mathbb{R}_{\geq 0}$ defined by $\lim_{M \to \infty} \frac{1}{M} \sum_{i=1}^M \delta_{\lambda_i^{K,p}}(\lambda)$ (convergence in distribution), where $\delta_{\lambda_i^{K,p}}$ is the Dirac measure.

**Empirical feature maps.** Consider $\mathcal{N}$ with $L$ layers. The mapping from an input $\mathbf{x}$ to $\mathbf{x}^{L-1}$ is a nonlinear transformation $\phi_\mathcal{N} : \mathbb{R}^{N_0} \mapsto \mathbb{R}^N$ called the empirical feature map, which is associated to an empirical kernel $K_\mathcal{N} : (\mathbf{x}, \mathbf{x}') \mapsto \langle \phi_\mathcal{N}(\mathbf{x}), \phi_\mathcal{N}(\mathbf{x}') \rangle$ expressed as the inner product between the corresponding feature map evaluations. By definition, $\mathcal{N}(\mathbf{x}) = (\mathbf{W}^L)^T \phi_\mathcal{N}(\mathbf{x})$ for any input $\mathbf{x}$. We always assume that each component of the feature map is Lipschitz continuous with respect to inputs $\mathbf{x}$, with a global Lipschitz constant noted $C$. This assumption holds for most commonly-used architectures including FCNs, CNNs and their variants, since the activation functions ReLU, sigmoid, tanh, as well as pooling and other commonly-used nonlinear operators are all Lipschitz continuous (Virmaux & Scaman, 2018).

Consider a training set $(\mathbf{X}, \mathbf{y})$ with $P > N$ examples sampled i.i.d. from an unknown distribution $p$ with compact support on $\mathbb{R}^{N_0} \times \mathbb{R}$. The evaluation $\phi_\mathcal{N}(\mathbf{X}) = (\phi_\mathcal{N}(\mathbf{x}_1), ..., \phi_\mathcal{N}(\mathbf{x}_P))^T \in \mathbb{R}^{N \times P}$ of the empirical feature map on the training set induces an *empirical feature covariance matrix* $\mathbf{K}_\mathcal{N}(\mathbf{X}, \mathbf{X}) \in \mathbb{R}^{P \times P}$, where element $(i, j)$ for $1 \leq i \leq j \leq N$ is given by $K_\mathcal{N}(\mathbf{x}_i, \mathbf{x}_j)$. Finally, the training data $\mathbf{X}$ also induces an *empirical probability distribution* $p_\mathbf{X}$ defined as $p_\mathbf{X}(\mathbf{x}) = \frac{1}{P} \left( \sum_{i=1}^P \delta_{\mathbf{x}_i}(\mathbf{x}) \right)$ with $\delta_{\mathbf{x}_i}$ the Dirac measure.

**Adversarial examples.** Consider a regression training set $(\mathbf{X}, \mathbf{y})$ sampled from an unknown distribution $p$ with compact support on $\mathbb{R}^{N_0} \times \mathbb{R}$ and consider $\mathcal{N}$ trained on $(\mathbf{X}, \mathbf{y})$ to zero error. Note that training to interpolation is a common assumption in deep learning theory (Belkin et al., 2019; Ishida et al., 2020; Belkin, 2021; Mallinar et al., 2022). Let $M_{\mathrm{adv}} > M_{\mathrm{nat}} > 0$ and $\epsilon > 0$. A vector $\mathbf{x}'$ is an $(\epsilon, M_{\mathrm{adv}}, M_{\mathrm{nat}})$-adversarial example for $\mathcal{N}$ and $p$ if there exists an example $(\mathbf{x}^*, y^*) \sim p$ such that $||\mathbf{x}^* - \mathbf{x}'|| \leq \epsilon$ and $|y^* - \mathcal{N}(\mathbf{x}^*)| \leq M_{\mathrm{nat}}$, but $|y^* - \mathcal{N}(\mathbf{x}')| \geq M_{\mathrm{adv}}$.

The prediction discrepancy factors $M_{\mathrm{adv}}, M_{\mathrm{nat}}$ are introduced to adjust the standard definition of adversarial example for classification to the regression setting, where there is no a-priori notion of what it means for the adversarial example to change the prediction (in classification, changing the prediction means predicting a different class). The prediction discrepancy factors $M_{\mathrm{adv}}, M_{\mathrm{nat}}$ allows practitioners to quantify when a modification of the output is significant for the regression task at hand, in particular they provide criteria for when an error is adversarial vs when an error is natural. To simplify the presentation, we fix arbitrary such $M_{\mathrm{adv}}, M_{\mathrm{nat}}$ and speak from now onwards of $\epsilon$-adversarial examples. The first condition in the definition requires that the adversarial example $\mathbf{x}'$ is close in norm to some test example for which $\mathcal{N}$ generalises to small error; the second requirement ensures that the corresponding error w.r.t. $\mathbf{x}'$ is large.

# 3 Adversarial Attacks Shift Mercer's Spectrum Towards Zero

In this section, we show that adversarial examples can be characterised as those that shift the Mercer's spectrum of the empirical kernel corresponding to a trained neural network to yield near-zero eigenvalues with sufficient probability.

Assume that $\mathcal{N}$, which has been trained (technically, to zero error) on a regression dataset $(\mathbf{X}, \mathbf{y})$ drawn from an unknown data distribution $p$. By Mercer's theorem, we have the existence of a unique collection of Mercer's eigenvalues $\lambda_i^{K_{\mathcal{N}}, p_{\mathbf{X}}}$ and eigenfunctions $\varphi_i^{K_{\mathcal{N}}, p_{\mathbf{X}}}$ associated to the empirical kernel $K_{\mathcal{N}}$ and the known empirical distribution $p_{\mathbf{X}}$ derived from the training data. Now, suppose that we sample a new example $(\mathbf{x}^*, y^*)$ using the true data distribution $p$ for which the $\mathcal{N}$ exhibits reasonable generalisation performance (technically, the error is smaller than the given $M_{\mathrm{nat}} > 0$), and consider any data point $\mathbf{x}'$ in the vicinity of $\mathbf{x}^*$. After augmenting the training set with the new example $\mathbf{x}'$, we reconsider the *updated* Mercer spectrum $\lambda_i^{K_{\mathcal{N}}, p_{\mathbf{X}:\mathbf{x}'}}$ for the same kernel $K_{\mathcal{N}}$ and the *updated* empirical distribution $p_{\mathbf{X}:\mathbf{x}'}$. In the next theorem, we show that $\mathbf{x}'$ is adversarial if and only if the eigenvalues in the updated Mercer spectrum exhibit sufficient density near zero. Consequently, the generation of adversarial attacks, using any known technique, can be conceptualised as a progression towards the eigenvalue space's zero point within the empirical kernel.

The main insight behind the proof is the observation that the local Lipschitz constant of $\mathcal{N}$ around $\mathbf{x}^*$, which quantifies the growth rate between the new data point $\mathbf{x}'$ and the true example $\mathbf{x}^*$ can be written as a sum of terms involving Mercer eigenvalues and eigenfunctions in the updated spectrum. In this decomposition, the value of terms depending on the eigenfunctions can be bounded as $\epsilon \to 0$, and hence the only way to have an adversarial local Lipschitz constant for small values of $\epsilon > 0$ is for the terms involving the eigenvalues to diverge to infinity.

**Theorem 3.1.** *Let $\mathcal{N}$ be a Lipschitz continuous neural network trained on $(\mathbf{X}, \mathbf{y}) \sim p$ to zero error. There exists $\epsilon_0 > 0$ such that for all $\epsilon \in ]0, \epsilon_0]$, the following equivalence holds true: a data point $\mathbf{x}' \in \mathbb{R}^{N_0}$ such that $||\mathbf{x}^* - \mathbf{x}'|| \leq \epsilon$ for some example $(\mathbf{x}^*, y^*) \sim p$ satisfying $(y^* - \mathcal{N}(\mathbf{x}^*))^2 \leq M_{\mathrm{nat}}$ is an $\epsilon$-adversarial example for $\mathcal{N}$ and $p$ if and only if the function $\lambda \mapsto \frac{1}{\lambda^2} \rho^{K_{\mathcal{N}}, p_{\mathbf{X}'}}(\lambda)$ is not integrable near zero [1], with $\mathbf{X}' = \mathbf{X}: \mathbf{x}'$.*

*Proof.* In this proof, for succinctness we denote matrix $\mathbf{K}_{\mathcal{N}}(\mathbf{X}, \mathbf{X})$ as $\mathbf{K}$. For $\epsilon > 0$ and $\mathbf{x}'$ an $\epsilon$-adversarial example for $\mathcal{N}$ and $p$, consider the local Lipschitz constant of $\mathcal{N}$:

$$A_{\mathbf{x}', \mathbf{x}^*} := \frac{|\mathcal{N}(\mathbf{x}') - \mathcal{N}(\mathbf{x}^*)|}{||\mathbf{x}' - \mathbf{x}^*||}$$

By the second triangular inequality, we have:

$$
\begin{aligned}
A_{\mathbf{x}', \mathbf{x}^*} \cdot \epsilon &\geq A_{\mathbf{x}', \mathbf{x}^*} \cdot ||\mathbf{x}' - \mathbf{x}^*|| \\
&= |\mathcal{N}(\mathbf{x}') - \mathcal{N}(\mathbf{x}^*)| \\
&\geq ||\mathcal{N}(\mathbf{x}') - y^*| - |\mathcal{N}(\mathbf{x}^*) - y^*|| \\
&\geq M_{\mathrm{adv}} - M_{\mathrm{nat}}
\end{aligned}
\tag{1}
$$

thus the local Lipschitz constant of $\mathcal{N}$ must be a diverging function of $\epsilon \to 0$ when $\mathbf{x}'$ is adverarial.

---

[1] That is, $\lim_{\substack{a \to 0^+ \\ a > 0}} \int_{\lambda = a}^{\infty} \frac{1}{\lambda^2} \rho^{K, p_{\mathbf{X}'}}(\lambda) \mathrm{d}\lambda = \infty$.

Since $\mathcal{N}$ interpolates $(\mathbf{X}, \mathbf{y})$ and $P > N$, the readout weights column matrix $\mathbf{W}^L$ of $\mathcal{N}$ is the unique solution of the linear system $[\phi_{\mathcal{N}}(\mathbf{X})]^T \mathbf{W}^L = \mathbf{y}$, which is given by the following expression involving the empirical feature map $\phi_{\mathcal{N}}$ and covariance matrix $\mathbf{K}$ of $\mathcal{N}$ on the training set $(\mathbf{X}, \mathbf{y})$: $\mathbf{W}^L = \phi_{\mathcal{N}}(\mathbf{X}) \mathbf{K}^\dagger \mathbf{y}$. Thus, the difference between evaluations of $\mathcal{N}$ on data points $\mathbf{x}', \mathbf{x}^*$ is given by the following expression:

$$\mathcal{N}(\mathbf{x}') - \mathcal{N}(\mathbf{x}^*) = [\phi_{\mathcal{N}}(\mathbf{x}') - \phi_{\mathcal{N}}(\mathbf{x}^*)]^T \left(\phi_{\mathcal{N}}(\mathbf{X})\mathbf{K}^\dagger \mathbf{y}\right) . \tag{2}$$

Dividing by $\|\mathbf{x}' - \mathbf{x}^*\|$, we have:

$$\frac{\mathcal{N}(\mathbf{x}') - \mathcal{N}(\mathbf{x}^*)}{\|\mathbf{x}' - \mathbf{x}^*\|} = \mathbf{c}_{\mathbf{x}',\mathbf{x}^*}^T \left(\phi_{\mathcal{N}}(\mathbf{X})\mathbf{K}^\dagger \mathbf{y}\right) . \tag{3}$$

The left hand side is equal to $\pm A_{\mathbf{x}',\mathbf{x}^*}$ depending on the sign of $\mathcal{N}(\mathbf{x}') - \mathcal{N}(\mathbf{x}^*)$ and on the right hand side, the vector $\mathbf{c}_{\mathbf{x}',\mathbf{x}^*}$ is such that each component $\pm(\mathbf{c}_{\mathbf{x}',\mathbf{x}^*})_i$ is the local Lipschitz constant of the $i$-th component of $\phi_{\mathcal{N}}$ (depending on the sign of $(\phi_{\mathcal{N}}(\mathbf{x}') - \phi_{\mathcal{N}}(\mathbf{x}^*))_i$). In particular each component verifies $|(\mathbf{c}_{\mathbf{x}',\mathbf{x}^*})_i| \leq C$. Combining equation 1 and equation 3, we have that the following term of interest $A_{\mathbf{x}',\mathbf{x}^*}^2 = \left(\mathbf{c}_{\mathbf{x}',\mathbf{x}^*}^T \left(\phi_{\mathcal{N}}(\mathbf{X})\mathbf{K}^\dagger \mathbf{y}\right)\right)^2$ diverges as a function of $\epsilon \to 0$ if $\mathbf{x}'$ is adversarial.

We note the vector $\mathbf{k} := \mathbf{c}_{\mathbf{x}',\mathbf{x}^*}^T \phi_{\mathcal{N}}(\mathbf{X})$ and calculate this term as:

$$\left(\mathbf{c}_{\mathbf{x}',\mathbf{x}^*}^T \left(\phi_{\mathcal{N}}(\mathbf{X})\mathbf{K}^\dagger \mathbf{y}\right)\right)^2 = \mathrm{Tr}\left(\mathbf{k}\mathbf{k}^T \mathbf{K}^\dagger \mathbf{y}\mathbf{y}^T \mathbf{K}^\dagger\right) \tag{4}$$

where we have used that $\mathbf{a}^T\mathbf{b} = \mathrm{Tr}(\mathbf{a}\mathbf{b}^T)$.

To conclude, let $\left(\lambda_i^{K_{\mathcal{N}}, p_{\mathbf{x}'}}, \varphi_i^{K_{\mathcal{N}}, p_{\mathbf{x}'}}\right)_{i \in \mathbb{N}}$ be the Mercer's decomposition of kernel $K_{\mathcal{N}}$ and the extended empirical distribution $p_{\mathbf{X}'}$. Then, Mercer's theorem provides an expression of the application of the kernel to data points in terms of the aforementioned eigenvalues and eigenfunctions; it follows that the empirical covariance matrix $\mathbf{K}$ can be written as follows for some large enough number $M \gg P$:

$$\mathbf{K} := \mathbf{\Phi}\mathbf{\Lambda}\mathbf{\Phi}^T \tag{5}$$

where $\mathbf{\Phi}_{j,k} = \varphi_k^{K_{\mathcal{N}}, p_{\mathbf{x}'}}(\mathbf{x}_j)$ for each $1 \leq j \leq P$ and $1 \leq k \leq M$ and $\mathbf{\Lambda}_{k,l} = \delta_{k,l}\lambda_k^{K_{\mathcal{N}}, p_{\mathbf{x}'}}$ for each $1 \leq k, l \leq M$.[2] Hence, using Mercer's decompositions, we can further expand equation 4 as follows:

$$\left(\mathbf{c}_{\mathbf{x}',\mathbf{x}^*}^T \left(\phi_{\mathcal{N}}(\mathbf{X})\mathbf{K}^\dagger \mathbf{y}\right)\right)^2 = \sum_j^M \frac{1}{(\lambda_j^{K_{\mathcal{N}}, p_{\mathbf{x}'}})^2} \cdot \left(\mathbf{\Phi}^\dagger \mathbf{y}\mathbf{y}^T (\mathbf{\Phi}^T)^\dagger\right)_{j,j} \left(\mathbf{\Phi}^\dagger \mathbf{k}\mathbf{k}^T (\mathbf{\Phi}^T)^\dagger\right)_{j,j}$$

$$+ \sum_j^M \sum_{k \neq j}^M \frac{1}{(\lambda_j^{K_{\mathcal{N}}, p_{\mathbf{x}'}})(\lambda_k^{K_{\mathcal{N}}, p_{\mathbf{x}'}})} \cdot$$
$$\left(\mathbf{\Phi}^\dagger \mathbf{y}\mathbf{y}^T (\mathbf{\Phi}^T)^\dagger\right)_{j,k} \left(\mathbf{\Phi}^\dagger \mathbf{k}\mathbf{k}^T (\mathbf{\Phi}^T)^\dagger\right)_{j,k}$$

With this expression at hand, we are ready to show the statement in the theorem. For the "if" direction, assume that $\mathbf{x}'$ is adversarial, then $\left(\mathbf{c}_{\mathbf{x}',\mathbf{x}^*}^T \left(\phi_{\mathcal{N}}(\mathbf{X})\mathbf{K}^\dagger \mathbf{y}\right)\right)^2 \to \infty$ as $\epsilon \to 0$.

The first step is to show that as $\epsilon \to 0$, the following quantities remain bounded: $\left(\mathbf{\Phi}^\dagger \mathbf{y}\mathbf{y}^T (\mathbf{\Phi}^T)^\dagger\right)_{j,j}$, $\left(\mathbf{\Phi}^\dagger \mathbf{k}\mathbf{k}^T (\mathbf{\Phi}^T)^\dagger\right)_{j,j}$, and $\left(\mathbf{\Phi}^\dagger \mathbf{y}\mathbf{y}^T (\mathbf{\Phi}^T)^\dagger\right)_{j,k}$, $\left(\mathbf{\Phi}^\dagger \mathbf{k}\mathbf{k}^T (\mathbf{\Phi}^T)^\dagger\right)_{j,k}$. Each of these can be written as sums over entries of the relevant matrices. In particular, since the rectangular matrix $\mathbf{\Phi} \in \mathbb{R}^{P \times M}$ has orthogonal rows, we have $\mathbf{\Phi}^\dagger = \mathbf{\Phi}^T \left(\mathbf{\Phi}\mathbf{\Phi}^T\right)^{-1}$, and $(\mathbf{\Phi}^T)^\dagger = \left(\mathbf{\Phi}\mathbf{\Phi}^T\right)^{-1} \mathbf{\Phi}$.

For the $i$-th entry of $\mathbf{k}$, we have $|\mathbf{k}_i| \leq \sqrt{N}C\|\phi_{\mathcal{N}}(\mathbf{x}_i)\|$ by Cauchy-Schwartz inequality. The entries $\mathbf{\Phi}$ remain bounded as $\epsilon \to 0$ as evaluations of Mercer's eigenfunctions. Similarly, the entries of $\mathbf{\Phi}\mathbf{\Phi}^T$ do not diverge as $\epsilon \to 0$, and neither do entries of $\left(\mathbf{\Phi}\mathbf{\Phi}^T\right)^{-1}$.

---

[2]Expression equation 5 is not the usual eigendecomposition of a square matrix: the evaluations of eigenfunctions yield rectangular (infinite) matrices. This decomposition is enabled by Mercer's theorem and applies to kernels.

Therefore, if the term of interest is to diverge towards infinity, at least one of the sums $\sum_j^M \frac{1}{(\lambda_j^{K_\mathcal{N}, p_{\mathbf{X}'}})^2}$, or $\sum_j^M \sum_{k \neq j}^M \frac{1}{(\lambda_j^{K_\mathcal{N}, p_{\mathbf{X}'}})(\lambda_k^{K_\mathcal{N}, p_{\mathbf{X}'}})}$ must diverge. For small eigenvalues, the first sum dominates and hence must diverge. This sum can be expressed using the density of Mercer's eigenvalues for $K_\mathcal{N}$ and $p_{\mathbf{X}'}$ as follows: $\lim_{M \to \infty} \sum_j^M \frac{1}{(\lambda_j^{K_\mathcal{N}, p_{\mathbf{X}'}})^2} = \int \frac{1}{\lambda^2} \rho^{K, p_{\mathbf{X}'}}(\lambda) \mathrm{d}\lambda$. Thus, the real-valued function $\lambda \mapsto \frac{1}{\lambda^2} \rho^{K, p_{\mathbf{X}'}}(\lambda)$ is not integrable near zero as required.

Conversely, assume that $\lambda \mapsto \frac{1}{\lambda^2} \rho^{K, p_{\mathbf{X}'}}(\lambda)$ is not integrable near zero. We have that, as $\epsilon \to 0$, $\left( \mathbf{\Phi}^\dagger \mathbf{y} \mathbf{y}^T (\mathbf{\Phi}^\dagger)^T \right)_{j,j} \left( \mathbf{\Phi}^\dagger \mathbf{k} \mathbf{k}^T (\mathbf{\Phi}^\dagger)^T \right)_{j,j}$ is bounded away from zero. Indeed, these terms can be written as sums of squared values, which can only converge to zero if each term of the sum converges to zero, which, in turn, only happens if $\mathbf{y} = 0$ or $\mathbf{k} = 0$ [3]. Thus, the term

$$\sum_j^M \frac{1}{(\lambda_j^{K_\mathcal{N}, p_{\mathbf{X}'}})^2} \left( \mathbf{\Phi}^\dagger \mathbf{y} \mathbf{y}^T (\mathbf{\Phi}^\dagger)^T \right)_{j,j} \cdot \left( \mathbf{\Phi}^\dagger \mathbf{k} \mathbf{k}^T (\mathbf{\Phi}^\dagger)^T \right)_{j,j}$$

causes a divergence in the local Lipschitz constant of $\mathcal{N}$ and example $\mathbf{x}'$ is adversarial, as required. □

It follows directly from the theorem that introducing $\mathbf{x}'$ yields a Mercer's decomposition where eigenvalues have sufficient density near zero. Indeed, if eigenvalues had insufficient density near zero then the function $\frac{1}{\lambda^2} \rho^{K_\mathcal{N}, p_{\mathbf{X}'}}(\lambda)$ would be integrable near zero. For instance, according to the convergence/divergence of Riemann integrals, $\rho^{K_\mathcal{N}, p_{\mathbf{X}'}}(\lambda) \sim \lambda^\beta$ as $\lambda \to 0$ leads to non-integrability if $\beta \leq 1$ and to integrability if $\beta > 1$. As a direct corollary of our result, if an example is close to a training or a test example and induces a substantial eigenvalue density near zero, then it is an adversarial example. This could inspire new defense mechanisms and new detection methods (Tramer, 2022); this is left for future work and we note that an important bottleneck, however, could be to quantify *a priori* whether a candidate example is close to a real (unobserved) example.

Our result is formally derived in the limit $\epsilon \to 0$. We would like to emphasize, however, that this assumption is needed to make a general statement about adversarial attacks. Indeed, small $\epsilon$ is implicit in the very definition of adversarial example, since it must be a small perturbation of a real example. In particular, the $\epsilon_0 > 0$, for which we have proved existence, relates to the adversarial robustness of a trained network: $\min_{\epsilon_0 > 0} \{ \exists \epsilon \in ]0, \epsilon_0] : \exists$ an $\epsilon -$ adversarial example$\}$ characterizes the adversarial example with smallest distance to a real example.

## 4 Adversarial Examples Are Exceedingly Unlikely

In this section, we exploit the result in Theorem 3.1 to show that adversarial examples have zero measure with respect to the true data distribution $p$. As in the previous section, we assume that the neural network is trained to zero error and that its generalisation error remains small for all examples drawn from the same distribution $p$ as the training data. In this setting, we prove that the probability of randomly sampling an $\epsilon$-adversarial example from $p$ tends to zero as $\epsilon \to 0$. Intuitively, Theorem 3.1 tells us that the density function $\frac{1}{\lambda^2} \rho^{K_\mathcal{N}, p_{\mathbf{X}}}(\lambda)$ for the training data is integrable near zero; this highly restricts the probability of sampling near-zero eigenvalues, and consequently also the probability of sampling adversarial examples.

**Theorem 4.1.** *Let $\mathcal{N}$ be a Lipschitz continuous neural network trained to zero error on $(\mathbf{X}, \mathbf{y}) \sim p$ and let $\mathbb{E}_{(\mathbf{x}, y) \sim p} \left( (y - \mathcal{N}(\mathbf{x}))^2 \right) \leq M_{\mathrm{nat}}$. Consider the indicator random variable $\mathbb{1}_\epsilon^\mathcal{N}$ which determines whether a vector $\mathbf{x} \sim p$ is an $\epsilon$-adversarial example for $\mathcal{N}$ and $p$. There exists $\epsilon_0 > 0$ such that for all $\epsilon \in ]0, \epsilon_0]$, it holds that $p(\mathbb{1}_\epsilon^\mathcal{N} = 1) = 0$.*

*Proof.* Before proving the statement of the theorem, we define a set of events with useful probabilities. For an arbitrary (but fixed) set $\mathbf{X}$ of data points sampled from $p$, let $\lambda_{\mathbf{X}}^j$ be the random variable assigning to each $\mathbf{x} \sim p$ the $j$-th Mercer eigenvalue for $K_\mathcal{N}$ and $p_{\mathbf{X} : \mathbf{x}}$. For an interval $[a, b]$, let $E_{[a,b]}^{\mathbf{X}, j}$ be the event of $\lambda_{\mathbf{X}}^j$ taking values within $[a, b]$ and define the event $\{\Lambda_{\mathbf{X}} = \lambda\} := \bigcup_j E_{[\lambda, \lambda + \mathrm{d}\lambda]}^{\mathbf{X}, j}$.

---

[3]Note that these excluded edge cases are already taken in account as particular cases of equation 2.

Let $\mathbb{1}_\epsilon^{\mathcal{N}}$ be the indicator random variable determining whether $\mathbf{x} \sim p$ is an $\epsilon$-adversarial example for $\mathcal{N}$ and $p$. By the law of total probability applied to $\mathbb{1}_\epsilon^{\mathcal{N}}$, the following holds:

$$p(\Lambda_{\mathbf{X}} = \lambda) = p(\Lambda_{\mathbf{X}} = \lambda | \mathbb{1}_\epsilon^{\mathcal{N}} = 1) \cdot p(\mathbb{1}_\epsilon^{\mathcal{N}} = 1) + p(\Lambda_{\mathbf{X}} = \lambda | \mathbb{1}_\epsilon^{\mathcal{N}} = 0) \cdot p(\mathbb{1}_\epsilon^{\mathcal{N}} = 0)$$

The probability $p(\Lambda_{\mathbf{X}} = \lambda)$ coincides with the average density of Mercer's eigenvalues with respect to the random vector $\mathbf{x} \sim p$:

$$p(\Lambda_{\mathbf{X}} = \lambda) = \int p(\Lambda_{\mathbf{X}} = \lambda | \mathbf{x}) p(\mathbf{x}) \mathrm{d}\mathbf{x} = \int \rho^{K_{\mathcal{N}}, p_{\mathbf{X}:\mathbf{x}}}(\lambda) p(\mathbf{x}) \mathrm{d}\mathbf{x}$$

Since $\mathbb{E}_{(\mathbf{x},y)\sim p(\mathbf{x},y)}\left((y - \mathcal{N}(\mathbf{x}))^2\right) \leq M_{\mathrm{nat}}$, it is easy to show, using the same arguments as in the proof of Theorem 1, that $\lambda \mapsto \frac{1}{\lambda^2} p(\Lambda_{\mathbf{X}} = \lambda)$ is integrable near zero.

By definition, $p(\Lambda_{\mathbf{X}} = \lambda | \mathbb{1}_\epsilon^{\mathcal{N}} = 1)$ is the average density of Mercer eigenvalues for $K_{\mathcal{N}}$ and $p_{\mathbf{X}:\mathbf{x}}$ given that $\mathbf{x}$ is an $\epsilon$-adversarial example. By Theorem 3.1, in the limit $\epsilon \to 0$, $p(\Lambda_{\mathbf{X}} = \lambda | \mathbb{1}_\epsilon^{\mathcal{N}} = 1)$ is strictly positive near zero, which gives us the following identity as $\lambda \to 0$ and $\epsilon \to 0$:

$$p(\mathbb{1}_\epsilon^{\mathcal{N}} = 1) = \frac{p(\Lambda_{\mathbf{X}} = \lambda)}{p(\Lambda_{\mathbf{X}} = \lambda | \mathbb{1}_\epsilon^{\mathcal{N}} = 1)} - \frac{p(\Lambda_{\mathbf{X}} = \lambda | \mathbb{1}_\epsilon^{\mathcal{N}} = 0)}{p(\Lambda_{\mathbf{X}} = \lambda | \mathbb{1}_\epsilon^{\mathcal{N}} = 1)} \cdot p(\mathbb{1}_\epsilon^{\mathcal{N}} = 0)$$

By Theorem 3.1, $\lambda \mapsto \frac{1}{\lambda^2} p(\Lambda_{\mathbf{X}} = \lambda | \mathbb{1}_\epsilon^{\mathcal{N}} = 1)$ is not integrable near-zero whereas $\lambda \mapsto \frac{1}{\lambda^2} p(\Lambda_{\mathbf{X}} = \lambda | \mathbb{1}_\epsilon^{\mathcal{N}} = 0)$ is integrable near-zero. In particular, $\frac{p(\Lambda_{\mathbf{X}} = \lambda | \mathbb{1}_\epsilon^{\mathcal{N}} = 0)}{p(\Lambda_{\mathbf{X}} = \lambda | \mathbb{1}_\epsilon^{\mathcal{N}} = 1)} \to 0$ as $\lambda \to 0$. Similarly, $\frac{p(\Lambda_{\mathbf{X}} = \lambda)}{p(\Lambda_{\mathbf{X}} = \lambda | \mathbb{1}_\epsilon^{\mathcal{N}} = 1)} \to 0$ as $\lambda \to 0$, which yields $p(\mathbb{1}_\epsilon^{\mathcal{N}} = 1) = 0$, as required. $\qquad\square$

Our findings indicate that in practical evaluations of neural models, it is highly improbable for test sets to include adversarial examples, given that they are sampled from the same underlying distribution $p$ as the training data. In essence, adversarial examples created through artificial perturbations of samples drawn from $p$ are considered out-of-distribution and thus extremely unlikely to originate from the same underlying data generation process that produced the training and test data. This substantiates the intuition that adversarial examples do not naturally manifest in real-world scenarios.

## 5 EXPERIMENTS

To validate our theory, we conducted experiments on a subset of MNIST consisting exclusively of classes "0" and "1", and a subset of CIFAR10 consisting exclusively of classes "plane" and "car". The subset of MNIST comprised 253 examples, each with 784 pixels per image. The subset of CIFAR10 comprised 100 examples each with 1024 coloured pixels. Additionally, we established a test set consisting of 17 examples (respectively, 10 examples) for MNIST (respectively, for CIFAR10). We deliberately opted for these reduced dataset sizes to accommodate computational constraints, as our computations require diagonalising kernel matrices with dimensions of $(P + 1) \times (P + 1)$. Furthermore, we used only two classes per experiment to conveniently turn classification tasks into one-dimensional regression tasks. However, it is important to note that the size of the dataset is inconsequential to our theoretical results, as they remain independent of dataset scale. All experiments were conducted on a GPU-enabled platform within Google Colab for enhanced computational efficiency.

Using Pytorch, we trained a ReLU FCN with one hidden layer of size $N = 512$ to zero error on our MNIST training dataset and $100\%$ accuracy on our restricted test set. Subsequently, we exploited the DeepFool algorithm (Moosavi-Dezfooli et al., 2016) to generate one adversarial example for each of the 17 test examples. This algorithm essentially involves an iterative process wherein we continuously adjust the input in the direction of the normalised gradient until a change in prediction occurs. On CIFAR10, we used the pre-trained network VGG (Simonyan & Zisserman, 2014) and the library torchattacks for their implementation of the Fast Adaptive Boundary (FAB) attack, which we used to generate an adversarial example for each of the 10 test examples, that are minimal perturbations with respect to the $\ell_\infty$ norm (Croce & Hein, 2020).

For each experiment, we computed the eigenvalue distributions of the empirical kernel for the original training data distribution. Then, for each adversarial example, we computed the updated eigenvalue distribution for the same empirical kernel and the training data extended with the adversarial

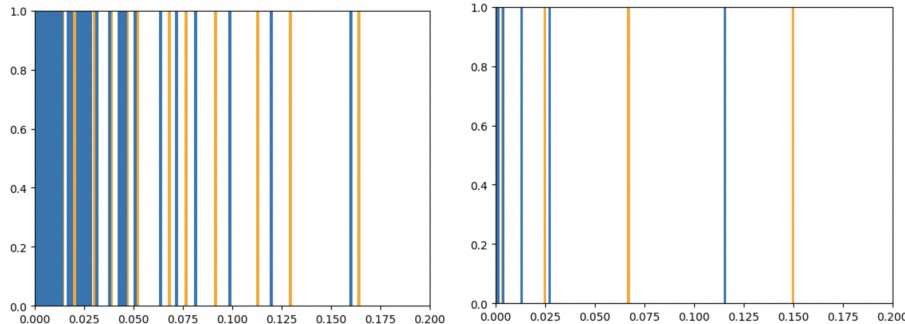

Figure 1: Eigenvalue distribution for kernel $K_{\mathcal{N}}$ and the original data distribution (respectively, the same data distribution extended with one adversarial example) in orange (respectively, in blue). The X-axis is indexed by the value of the eigenvalues and the Y-axis is a probability density. On the left, the result for the MNIST experiment and on the right, the result for the CIFAR10 experiment.

example. Thus, this gives us 17 different updated eigenvalue distributions for the MNIST experiment and 10 different updated eigenvalue distributions for the CIFAR10 experiment. To compute these distributions, we constructed 17 (respectively 10) new kernel matrices by adding the relevant row and column corresponding to the adversarial example and diagonalised them using PyTorch. By the classical results of Baker (1977); Rasmussen & Williams (2006), the largest Mercer eigenvalues roughly coincide with the eigenvalues obtained by diagonalising the corresponding kernel matrix. We have plotted for each experiment examples of such updated distributions against the original ones in Figure 1; note how introducing an adversarial example shifts the eigenvalue distribution towards zero.

We computed the minimal eigenvalue within each eigenvalue distribution, and we estimated the integral near zero of our real-valued function $\lambda \mapsto \frac{1}{\lambda^2}\rho(\lambda)$ by computing the sum $\frac{1}{B}\sum_j \frac{1}{\lambda_j^2}\rho(\lambda_j)$ for the relevant densities of Mercer eigenvalues over the $B$ bins in the histograms of Figure 1. The histograms representing these estimations are depicted in Figure 2. Our numerical results clearly demonstrate that, as expected, introducing adversarial examples shifts the eigenvalue distribution towards zero and inflates the value of the integral of interest.

Additional experiments, including with other well-known attack methods, are provided in the appendix. The results are largely consistent across experiments.

## 6    RELATED WORK

Several explanations have been proposed for the pervasiveness of adversarial attacks in neural networks. The *linearity hypothesis* (Goodfellow et al., 2015) posits that, because of the (local) linear nature of trained neural networks, small changes to each component of a high-dimensional vector amount to a large change in the network's output. The universality of $\ell_2$-adversarial attacks on ReLU networks with random weights (Daniely & Shacham, 2020) provides a strong argument supporting this hypothesis, especially considering the locally linear characteristics of ReLU networks. The linearity hypothesis has motivated a series of research endeavors analysing the topological characteristics of decision boundaries, with the aim to elucidate the nature of adversarial examples and develop techniques for enhancing adversarial robustness. Tanay & Griffin (2016) argues that the linearity hypothesis is not fully satisfactory, and rather attributes adversarial examples to the distance of the sampling subspace to the decision boundary. In other related works, such as Simon-Gabriel et al. (2019), the susceptibility to adversarial attacks escalates with the increase in input dimensionality. Deficiencies in the topology become more pronounced in higher dimensions, a phenomenon exacerbated by the curse of dimensionality. In some instances, particularly for synthetic data distributions with sufficiently high dimensions, adversarial attacks become nearly unavoidable, as noted in Shafahi et al. (2019). Moreover, data sparsity in relation to the input space heightens vulnerability to adversarial attacks, as discussed in Paknezhad et al. (2022); Weitzner & Giryes (2023). The dimensionality of the parameter space also plays a significant role in this context, with param-

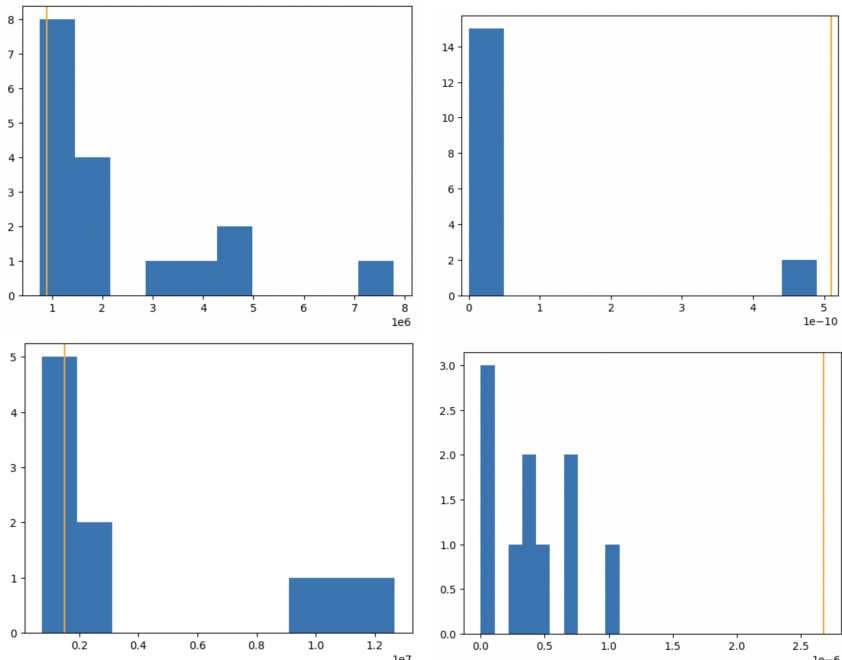

Figure 2: Comparing the integral of interest (left subfigures) and the minimal eigenvalue (right subfigures) between the original data distribution (orange line) and the updated data distributions (in blue). The X-axis is indexed by the value of the integral quantity (respectively, the minimal value of eigenvalues) on the left (respectively, on the right). In turn, the Y-axis is indexed on the left (respectively, on the right) by the number of updated data distributions whose estimated integral (respectively, whose minimal eigenvalue) falls into that bin. On the top, the results for the MNIST experiment and on the bottom, the results for the CIFAR10 experiment.

eter redundancy (Paknezhad et al., 2022) and a large $\ell_1$-norm of the parameters (Guo et al., 2018) representing situations associated with increased vulnerability.

Another line of research identifies features as the key object driving the occurrence of adversarial examples. In Ilyas et al. (2019), it was shown that *non-robust features* (those resulting from spurious data correlations that are nevertheless highly predictive) are responsible for the presence of adversarial examples. Similarly, Wang et al. (2017) established that the existence of an unnecessary feature introduced to replicate the true underlying target function renders a system vulnerable to adversarial attacks. This perspective is further supported by the observation that saliency methods tend to emphasise class-discriminative features which can be exploited by attackers Gu & Tresp (2019).

Our approach is positioned at the intersection of both these streams of research. By taking Mercer's eigenvalues into account, our theory operates in the high-dimensional embedding space where the predictions follow a linear pattern. Additionally, shifting towards zero eigenvalues can be understood as exploiting directions of non-robust features. Indeed, our approach was first inspired by the double-descent phenomenon in neural networks (Mei & Montanari, 2022), in which the spectrum of the empirical kernel also shifts towards zero at the divergence in terms of generalisation error. Within this body of literature, it is already established that stochastic cancellations give rise to the emergence of spurious directions within the feature space (El Harzli et al., 2024), causing them to overfit.

A final perspective, which bears loose connections with our approach, is based on information geometry (Zhao et al., 2019; Naddeo et al., 2022). This perspective centers on the use of the *Fisher Information Matrix* which quantifies, for any pair of data points, the correlation between parameter gradients of the log likelihood of the data. A technique involving the iterative elimination of the dominant eigenvalue direction in the Fisher Information Matrix leads to the generation of adversarial examples. This process appears to yield configurations where variations in network parameters

exhibit strong linear dependencies with respect to the data. These configurations may manifest whenever certain features become perfectly correlated within the dataset and this, in turn, results in the empirical kernel having Mercer eigenvalues that approach zero.

## 7 LIMITATIONS AND FUTURE WORK

Our results are currently directly applicable to one-dimensional regression tasks, but we anticipate that our core insights could be generalised to encompass classification and multidimensional regression. Additionally, conducting comprehensive experiments on more extensive and diverse datasets would be beneficial. Importantly, our findings remain unaffected by variations in problem dimensions and data characteristics, offering a high degree of versatility.

Our results are derived in the limit where adversarial examples become infinitesimally close to data points. Specifically, we have not characterised the rate at which the probability of encountering adversarial examples diminishes with the distance to an example from the true data distribution. This remains a subject for future research and requires the derivation of precise bounds for the generalisation error. Exploring specific criteria governing the data distribution and neural architecture that lead to rapid convergence rates (as $\epsilon \to 0$) and result in low probabilities of generating adversarial examples is a promising avenue. Achieving this understanding could lay the groundwork for designing more robust architectures. Additionally, while we have proved that adversarial examples have a measure zero with respect to the data distribution in the limit, we have not shown that examples producing near-zero Mercer eigenvalues for the empirical kernel of a neural network always exist.

In conclusion, we anticipate that our findings will serve as a catalyst for research into the characteristics and prevalence of adversarial examples. Furthermore, we hope to encourage the exploration and development of novel defense mechanisms and detection methods against adversarial attacks.

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

# A  ADDITIONAL EXPERIMENTS

In this section, we present additional experimental results for completeness. We tested different combinations of attack methods and tasks. In particular, we experimented with two other well-known attack methods: Projected Gradient Descent (PGD) Madry et al. (2018) and Fast Gradient Sign Method (FGSM) Goodfellow et al. (2015). The results are consistent across experiments. Finally, we augmented the number of data points for one experiment to showcase that the distributions remain consistent as the number of examples increases.

## A.1  REMAINING EXPERIMENTS FOR DEEPFOOL AND FAB ATTACK

We conducted the same experiments as described in the main text but swapping DeepFool and FAB attack.

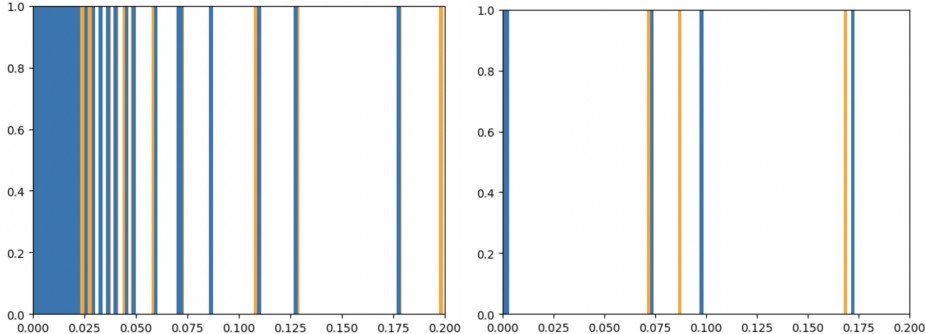

Figure 3: Eigenvalue distribution for kernel $K_\mathcal{N}$ and the original data distribution (respectively, the same data distribution extended with one adversarial example) in orange (respectively, in blue). The X-axis is indexed by the value of the eigenvalues and the Y-axis is a probability density. On the left, the result for the MNIST experiment with FAB attack and on the right, the result for the CIFAR10 experiment with DeepFool.

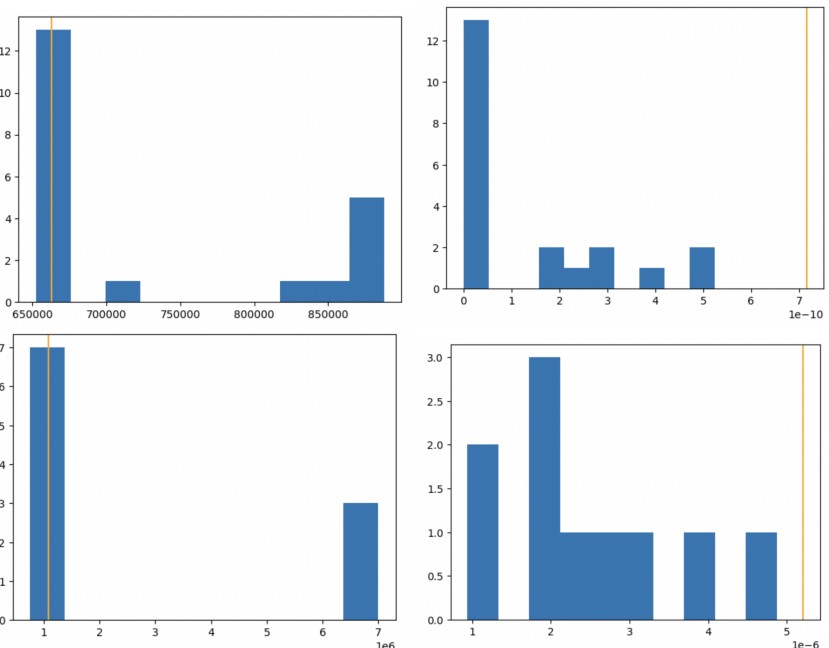

Figure 4: Comparing the integral of interest (left subfigures) and the minimal eigenvalue (right subfigures) between the original data distribution (orange line) and the updated data distributions (in blue). The X-axis is indexed by the value of the integral quantity (respectively, the minimal value of eigenvalues) on the left (respectively, on the right). In turn, the Y-axis is indexed on the left (respectively, on the right) by the number of updated data distributions whose estimated integral (respectively, whose minimal eigenvalue) falls into that bin. On the top, the results for the MNIST experiment with FAB attack and on the bottom, the results for the CIFAR10 experiment with Deep-Fool.

## A.2  FGSM

We conducted the same experiments as described in the main text with the FGSM attack.

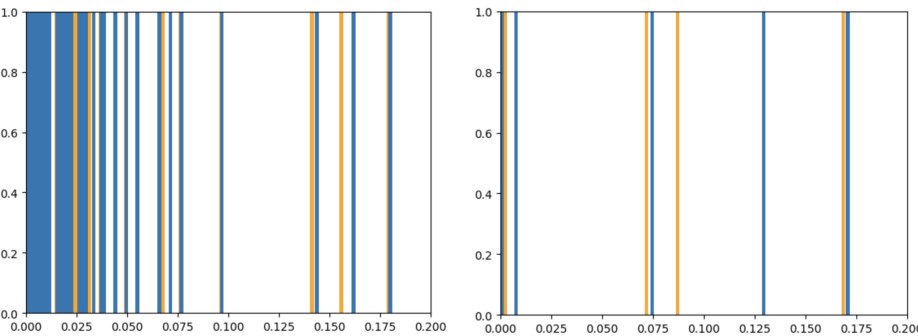

Figure 5: Eigenvalue distribution for kernel $K_{\mathcal{N}}$ and the original data distribution (respectively, the same data distribution extended with one adversarial example) in orange (respectively, in blue). The X-axis is indexed by the value of the eigenvalues and the Y-axis is a probability density. On the left, the result for the MNIST experiment with FGSM attack and on the right, the result for the CIFAR10 experiment with FGSM.

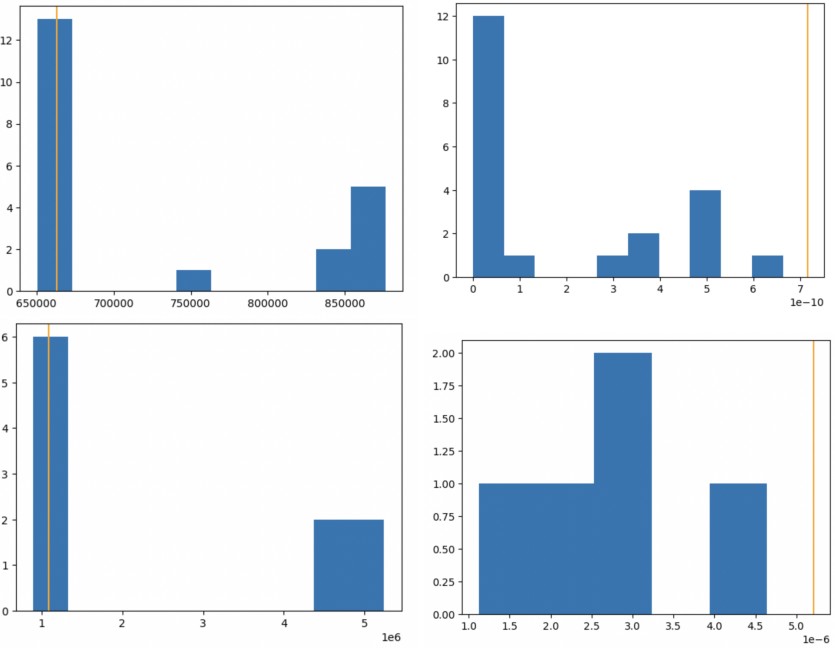

Figure 6: Comparing the integral of interest (left subfigures) and the minimal eigenvalue (right subfigures) between the original data distribution (orange line) and the updated data distributions (in blue). The X-axis is indexed by the value of the integral quantity (respectively, the minimal value of eigenvalues) on the left (respectively, on the right). In turn, the Y-axis is indexed on the left (respectively, on the right) by the number of updated data distributions whose estimated integral (respectively, whose minimal eigenvalue) falls into that bin. On the top, the results for the MNIST experiment with FGSM attack and on the bottom, the results for the CIFAR10 experiment with FGSM.

## A.3 PGD

We conducted the same experiments as described in the main text with the PGD attack.

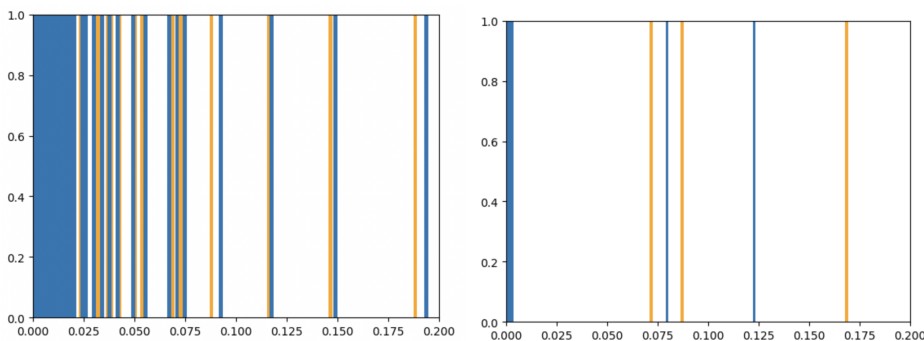

Figure 7: Eigenvalue distribution for kernel $K_{\mathcal{N}}$ and the original data distribution (respectively, the same data distribution extended with one adversarial example) in orange (respectively, in blue). The X-axis is indexed by the value of the eigenvalues and the Y-axis is a probability density. On the left, the result for the MNIST experiment with PGD attack and on the right, the result for the CIFAR10 experiment with PGD.

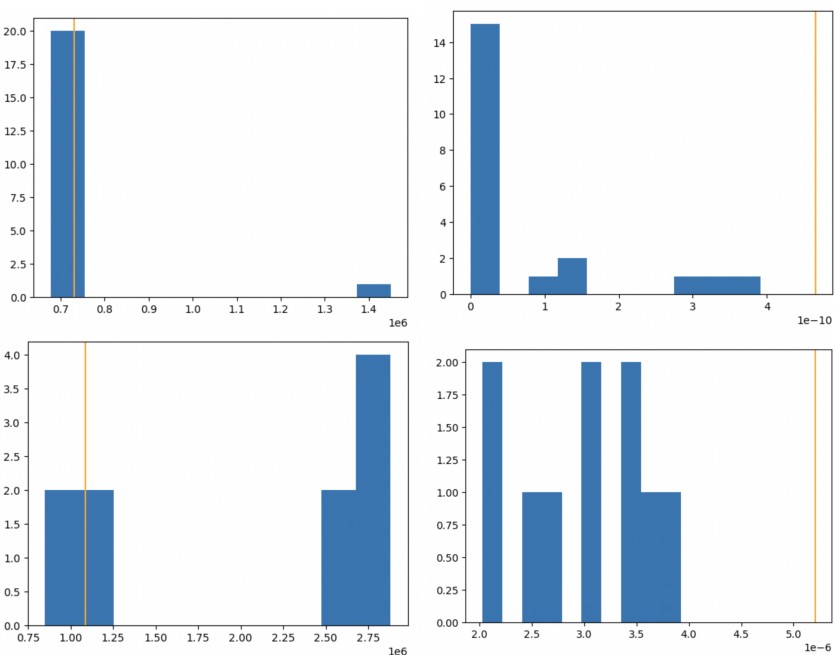

Figure 8: Comparing the integral of interest (left subfigures) and the minimal eigenvalue (right subfigures) between the original data distribution (orange line) and the updated data distributions (in blue). The X-axis is indexed by the value of the integral quantity (respectively, the minimal value of eigenvalues) on the left (respectively, on the right). In turn, the Y-axis is indexed on the left (respectively, on the right) by the number of updated data distributions whose estimated integral (respectively, whose minimal eigenvalue) falls into that bin. On the top, the results for the MNIST experiment with PGD attack and on the bottom, the results for the CIFAR10 experiment with PGD.

## A.4 PGD with more examples

For this set of experiments, we also increased the number of examples to 500 training points and 100 test points. The distributions remain consistent with the smaller dataset.

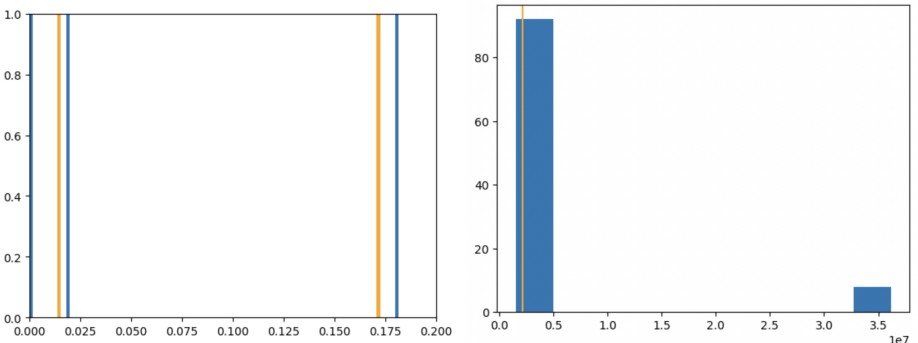

Figure 9: On the left, eigenvalue distribution for kernel $K_{\mathcal{N}}$ and the original data distribution (respectively, the same data distribution extended with one adversarial example) in orange (respectively, in blue). The X-axis is indexed by the value of the eigenvalues and the Y-axis is a probability density. On the right, comparing the integral of interest between the original data distribution (orange line) and the updated data distributions (in blue). The X-axis is indexed by the value of the integral quantity and the Y-axis is indexed by the number of updated data distributions whose estimated integral falls into that bin.