# OpenReview forum: "Adversarial Attacks as Near-Zero Eigenvalues in the Empirical Kernel of Neural Networks"
_ICLR.cc/2025/Conference — Submitted to ICLR 2025_

### Official Review · Reviewer_8xgg · 2024-10-21

**Soundness:** 3
**Presentation:** 3
**Contribution:** 2
**Rating:** 5
**Confidence:** 3

**Summary:**

This article mainly uses Mercer theorem to provide two theorems about the adversarial samples. The Theorem 3.1 shows that $x'$ is an adversarial exampls if and only if the density of Mercer’s eigenvalues which based on the network and $x'$ is not integrable near zero. The Theorem 4.1  shows that adversarial examples are zero measure. And last, author gives some experiments.

**Strengths:**

1, This article attempts to understand adversarial samples from the perspective of Kernel functions, and demonstrates the characteristics (Th3.1) and density (Th4.1) of adversarial samples. These two research questions are undoubtedly important.

2, The proof of the theorem is clear.

3, Nice write and easy to follow.

**Weaknesses:**

1, In my opinion, Theorem 3.1 seems to provide a method for determining whether a new sample is an adversarial sample by using the feature vectors of the samples in the training set and the new sample. Moreover, this method seems quite cumbersome, I couldn't see the importance of it.

2, Th4.1 seems obvious. The author's definition of adversarial samples is a sample that is misclassified near a correctly classified sample (line 161), and the network is Lip continuous, so there should be a small neighborhood around each correctly classified samples so that the points in the neighborhood are naturally correct. From this perspective, theorem 4.1 is clearly established.

**Questions:**

My biggest question about this article is the importance of the results given by the two theorems.

---

> ### Author Response · Authors · 2024-11-19
>
> Comment: Theorem 3.1 seems to provide a method for determining whether a new sample is an adversarial sample by using the feature vectors of the samples in the training set and the new sample.
>
> Answer: As noted in our response to Reviewers EDzP and azHL, our paper introduces a novel characterisation of adversarial examples within the kernel space, in contrast to the typical focus on the feature space in existing literature (see lines 463–472 in our initial submission and lines 466-475 in our revised submission). Our novel characterisation opens the way for the future use of well-established techniques from kernel methods in this context.
>
> Comment: Th4.1 seems obvious. The author's definition of adversarial samples is a sample that is misclassified near a correctly classified sample (line 161), and the network is Lip continuous, so there should be a small neighborhood around each correctly classified samples so that the points in the neighborhood are naturally correct. From this perspective, theorem 4.1 is clearly established.
>
> Answer: Theorem 4.1 assumes that the network outputs correct predictions *in expectation* (with respect to the data distribution) so there is no obvious reason to believe that there can't be a substantial probability to sample an incorrect example from the distribution. Please note that Theorem 4.1 is a global result (with respect to the entire data distribution) rather than a local result (which would be stated for each specific example).

---

> > ### Comment · Reviewer_8xgg · 2024-11-22
> >
> > My biggest question is what I said: 'Moreover, this method seems quite cumbersome, I couldn't see the importance of it.'.
> >
> > Authors answer that: Our novel characterisation opens the way for the future use of well-established techniques from kernel methods in this context. And in the paper, authors said  'Our approach is positioned at the intersection of both these streams of research.' and 'our approach was first inspired by the double-descent phenomenon in neural networks'.
> >
> > I think these answers lack some key points. I would like to see more specific: For the method of detecting adversarial samples in th3.1, what problems does this method help solve? What research fields is this method important for?

---

> > > ### Author Response · Authors · 2024-11-22
> > >
> > > Apologies for this, our response "our paper introduces a novel characterisation of adversarial examples within the kernel space, in contrast to the typical focus on the feature space in existing literature (see lines 463–472 in our initial submission and lines 466-475 in our revised submission). Our novel characterisation opens the way for the future use of well-established techniques from kernel methods in this context." was indeed for both your question about Theorem 3.1 and your bigger question on the importance of the results.
> > >
> > > More specifically, for detection of adversarial examples, as explained in our response to Reviewer EDzP, our results could inspire detection methods for adversarial examples involving identifying data points that yield near-zero eigenvalues, and this is consistent with Tramer (2022)'s result that shows that detecting adversarial examples is almost as hard as classifying them. Indeed, assuming that we have identified an example that produce near-zero eigenvalues, to have a detection of an adversarial example, it remains to determine whether the example is close in distance to a true (potentially unobserved) example: this is as hard as classifying the example correctly.
> > >
> > > Our results contribute new insights to the field of the theory of adversarial examples, whose aim is to understand their nature and their behavior, and more generally to learning theory of neural networks. Our results readily help understanding the rarity of naturally-sampled adversarial examples (Theorem 4.1), and could help understanding the pervasiveness of artificially-crafted adversarial examples (by studying to what extent, given data and architecture, it is possible to construct an example that yield near-zero eigenvalues in the empirical kernel). We argued in our paper that our results are positioned at the intersection of two important theories of adversarial examples: the linearity hypothesis and the non-robust features hypothesis. Furthermore, our results establish a new connection with another phenomenon studied in learning theory, the double-descent phenomenon, where near-zero eigenvalues are also the main driver of the phenomenon. Intuitively, this makes sense because both phenomena (adversarial examples and double descent) pertain to the generalisation error of neural networks and correspond to pathological configurations.

---

> > > > ### Comment · Reviewer_8xgg · 2024-11-25
> > > >
> > > > Can the author provide some practical scenarios and experiments to illustrate the important role of this methodin th3.1 in certain fields? Or is the main contribution of this method to inspire future research ideas?

---

> > > > > ### Author Response · Authors · 2024-11-25
> > > > >
> > > > > While the primary contribution of our work is indeed conceptual/theoretical, our characterisation of adversarial examples could pave the way for exploring novel methods to detect such examples and assess neural network robustness. Specifically, as a practical scenario, our characterisation ensures that any example that is close in distance to a true example  and which yields near-zero eigenvalues in the network's empirical kernel qualifies as an adversarial example for that network. However, this insight alone is insufficient for devising practical algorithms due to the inherent challenge of determining whether an example is close to a true example (detecting adversarial examples is as hard as classifying them). We hope that this new understanding of these practical hurdles will stimulate further research.

---

> > > > > > ### Comment · Reviewer_8xgg · 2024-11-26
> > > > > >
> > > > > > I acknowledge the author's contribution in theory, but the practicality of the algorithm still concerns me. So I will maintain my rating.

---

### Official Review · Reviewer_azHL · 2024-10-31

**Soundness:** 3
**Presentation:** 2
**Contribution:** 2
**Rating:** 6
**Confidence:** 3

**Summary:**

This paper provides one theoretical result that substantiate the original hypothesis raised in Szegedy's 2014 paper. This research paper is based on the formulation of NN as specific instances of kernel machines, and it adopts the perspective that the introduction of an adversarial example can be regarded as a modification of the empirical data distribution derived from the training set, expanded to incorporate the adversarial example.

The results include a mathematical explanation for that while NN demonstrate strong generalisation to novel test examples, they are still susceptible to attacks, because adversarial examples have measure zero in the limit where they become infinitely close to any test example sampled from the original data distribution. There also numerical experiments conducted that align with the theory.

**Strengths:**

This paper provides a relatively new perspective on explaining adversarial examples in NNs.
There is an explainable math framework.

**Weaknesses:**

The paper made a strong assumption that the adversarial examples are close to the data points.

Experiments are relatively insufficient, for example, there are only experiments over two specific classes of both MNIST and CIFAR-10 datasets.

**Questions:**

Is there any numerical/theoretical comparison between this theory and other explanations of the frangibility of NN networks? Why do you think this paper has a better interpretation?

---

> ### Author Response · Authors · 2024-11-19
>
> Comment: Experiments are relatively insufficient, for example, there are only experiments over two specific classes of both MNIST and CIFAR-10 datasets.
>
>  Answer: As pointed out in the response to Reviewer EDzP, we have included additional experiments in the appendix, incorporating FGSM and PGD attacks as well as testing on different combinations of attacks and tasks. Please refer to Figures 3-9 in our revised submission for details.
>
> Question: Is there any numerical/theoretical comparison between this theory and other explanations of the frangibility of NN networks? Why do you think this paper has a better interpretation?
>
> Answer: We  have provided theoretical comparisons with other interpretations (l. 470-485 in our initial submission and l. 473-488 in our revised submission). As noted in our response to Reviewer EDzP, our paper introduces a novel characterisation of adversarial examples within the kernel space, providing a direct explanation for the rarity of adversarial examples and paving the way for the future use of well-established techniques from kernel methods in this context.

---

### Official Review · Reviewer_EDzP · 2024-11-04

**Soundness:** 3
**Presentation:** 2
**Contribution:** 2
**Rating:** 5
**Confidence:** 3

**Summary:**

This paper proposes a new theory explaining the nature of adversarial examples. Using Mercer’s theorem from kernel theory, the authors try to explain the adversarial examples’ contradictory nature; even a neural network with good generalization performance suffers from the existence of adversarial examples. First, the authors provide a theorem that adversarial examples shift Mercer’s spectrum to have near-zero eigenvalues. Then, the authors prove another theorem that explains why it is unlikely to sample an adversarial example in a test set. Finally, the authors validate their theory with a set of experiments.

**Strengths:**

1. The theoretical finding is non-trivial. The proofs look to be correct.
2. The paper proposes a new theoretical analysis of adversarial examples. To the best of my knowledge, this is the first application of kernel theory, and the authors make a novel contribution to the theoretical understanding of adversarial examples.
3. The theoretical finding does not depend on a specific data point being attacked.
4. The authors tried validating the theory on a more complex neural network architecture, i.e., VGG.

**Weaknesses:**

1. The significance of the theoretical findings is unclear. The authors should have demonstrated more potential application of the theoretical findings, e.g., how to apply the findings to design a new defense.
2. The theoretical finding is a limit behavior that only holds for extremely small perturbation sizes. The authors’ argument that such an assumption on small perturbation size is needed is not sufficient because we don’t even have a proper definition of how small is “small enough”. As the authors describe in Section 7, a formal convergence rate analysis is required.
3. The experiments should be improved further.
   * The number of adversarial examples is too small to demonstrate some distribution.
   * The experiment setup is quite arbitrary. Different architectures and attacks are used for different datasets.
   * The choice of attack algorithms (DeepFool and FAB) also seems uncommon. It is better to use simpler and common attack algorithms.

**Questions:**

1. Can the authors provide more evidence about the significance of the theoretical findings? You may refer to more existing papers that reflect the findings, or you can provide a more detailed impact statement.
   * I can see that the authors mentioned (Tramer, 2022) as a detection method that the theoretical findings could inspire. Can the authors provide more details about how this detection method relates to your findings?
2. In my opinion, characterization of the convergence rate is a necessary part that completes the paper. Statements that hold only at the limit are too weak to be accepted.
3. Comments regarding the experiments.
   * I understand that the experiments require enormous computation, but the number of adversarial examples is too small to say anything about the distribution.
   * Do you have some specific reason for using DeepFool or FAB to generate adversarial examples? If the experiments take a long time, why don’t you try simpler attacks, such as FGSM or PGD with a small number of iterations?
   * Why do you use different network architectures and attack methods for each dataset? If you have experimental results for all the other combinations (e.g., VGG architecture on MNIST, DeepFool attack on CIFAR-10, etc.), you may put them in the Appendix and refer to the Appendix. If you don’t provide those results, it looks like you are cherry-picking the best combinations and hiding some negative results.

---

> ### Author Response · Authors · 2024-11-19
>
> Comment: "As the authors describe in Section 7, a formal convergence rate analysis is required."
>
> Answer: Characterising $\epsilon$-adversarial attacks for $\epsilon > 0$ without taking the limit (for example with a bound) is comparable to the challenge of characterising the generalisation error of neural networks---a very hard open problem. Our work can be seen as a step in this direction.
>
> Question: I understand that the experiments require enormous computation, but the number of adversarial examples is too small to say anything about the distribution.
>
> Answer: We have increased the number of examples (500 training examples and 100 test examples) in one of our experiments to show that the distributions remain consistent as the number of examples increases. Please see Figure 9 in our revised submission.
>
> Question: Can the authors provide more evidence about the significance of the theoretical findings? You may refer to more existing papers that reflect the findings, or you can provide a more detailed impact statement.
>
> Answer: Theorem 3.1 characterises the occurrence of adversarial examples within the kernel space, in contrast to the typical focus on the feature space in existing literature (see lines 463–472 in our initial submission and lines 466-475 in our revised submission). Our work demonstrates that this kernel space characterisation yields novel results, including a direct explanation for the rarity of adversarial examples. We anticipate that exploring the kernel space further could offer valuable new insights in the future.
>
> Comment: I can see that the authors mentioned (Tramer, 2022) as a detection method that the theoretical findings could inspire. Can the authors provide more details about how this detection method relates to your findings?
>
> Answer: Tramer (2022) demonstrate that detecting adversarial examples is almost as hard as their classification, i.e., there exists an algorithm (which preserves sample complexity) for constructing a robust classifier from a robust detector. Our results suggest detection methods for adversarial examples involving identifying data points that yield near-zero eigenvalues, and this is consistent with Tramer's result. Indeed, assume that we have identified an example that produce near-zero eigenvalues. To have a detection of an adversarial example, it remains to determine whether the example is close in distance to a true (potentially unobserved) example: this is as hard as classifying the example correctly.
>
> Comments: "Do you have some specific reason for using DeepFool or FAB to generate adversarial examples? If the experiments take a long time, why don’t you try simpler attacks, such as FGSM or PGD with a small number of iterations?" and
> "Why do you use different network architectures and attack methods for each dataset? If you have experimental results for all the other combinations (e.g., VGG architecture on MNIST, DeepFool attack on CIFAR-10, etc.), you may put them in the Appendix and refer to the Appendix."
>
> Answer: Our approach is method-agnostic, and we initially selected DeepFool and FAB as representative adversarial attack techniques. We believed that our experiments would provide sufficient variation to ensure the consistency of our results was unlikely to be coincidental. However, in response to feedback, we have included additional experiments in the appendix, incorporating FGSM and PGD attacks as well as testing on different combinations. Please refer to Figures 3-9 in our revised submission for details.

---

> > ### Comment · Reviewer_EDzP · 2024-11-27
> >
> > ### Further theoretical analysis
> > > Comment: "As the authors describe in Section 7, a formal convergence rate analysis is required."
> > >
> > > Answer: Characterising $\epsilon$-adversarial attacks for $\epsilon>0$ without taking the limit (for example with a bound) is comparable to the challenge of characterising the generalisation error of neural networks---a very hard open problem. Our work can be seen as a step in this direction.
> >
> > I agree that showing this theoretically is a challenging problem. Then, this paper should demonstrate its practical value. This does not mean the authors should design a new detection/defense based on the theoretical finding. However, the authors should demonstrate that,
> >
> > 1. The finding reflects the reality well (I think that the authors mainly focus on this),
> > 2. The finding either reflects the existing detection methods, e.g., all the detected adversarial examples satisfy the proved property, or can improve those detection methods, e.g., there are undetected adversarial examples that still satisfy the property
> >
> > ### More experiments
> > > Question: I understand that the experiments require enormous computation, but the number of adversarial examples is too small to say anything about the distribution.
> > >
> > > Answer: We have increased the number of examples (500 training examples and 100 test examples) in one of our experiments to show that the distributions remain consistent as the number of examples increases. Please see Figure 9 in our revised submission.
> >
> > > Comments: "Do you have some specific reason for using DeepFool or FAB to generate adversarial examples? If the experiments take a long time, why don’t you try simpler attacks, such as FGSM or PGD with a small number of iterations?" and "Why do you use different network architectures and attack methods for each dataset? If you have experimental results for all the other combinations (e.g., VGG architecture on MNIST, DeepFool attack on CIFAR-10, etc.), you may put them in the Appendix and refer to the Appendix."
> > >
> > > Answer: Our approach is method-agnostic, and we initially selected DeepFool and FAB as representative adversarial attack techniques. We believed that our experiments would provide sufficient variation to ensure the consistency of our results was unlikely to be coincidental. However, in response to feedback, we have included additional experiments in the appendix, incorporating FGSM and PGD attacks as well as testing on different combinations. Please refer to Figures 3-9 in our revised submission for details.
> >
> > I appreciate the additional experiments and believe the added experimental results strengthened the paper.

---

> > > ### Comment · Reviewer_EDzP · 2024-11-27
> > >
> > > ### Importance of the findings
> > > > Question: Can the authors provide more evidence about the significance of the theoretical findings? You may refer to more existing papers that reflect the findings, or you can provide a more detailed impact statement.
> > > >
> > > > Answer: Theorem 3.1 characterises the occurrence of adversarial examples within the kernel space, in contrast to the typical focus on the feature space in existing literature (see lines 463–472 in our initial submission and lines 466-475 in our revised submission). Our work demonstrates that this kernel space characterisation yields novel results, including a direct explanation for the rarity of adversarial examples. We anticipate that exploring the kernel space further could offer valuable new insights in the future.
> > >
> > > > Comment: I can see that the authors mentioned (Tramer, 2022) as a detection method that the theoretical findings could inspire. Can the authors provide more details about how this detection method relates to your findings?
> > > >
> > > > Answer: Tramer (2022) demonstrate that detecting adversarial examples is almost as hard as their classification, i.e., there exists an algorithm (which preserves sample complexity) for constructing a robust classifier from a robust detector. Our results suggest detection methods for adversarial examples involving identifying data points that yield near-zero eigenvalues, and this is consistent with Tramer's result. Indeed, assume that we have identified an example that produce near-zero eigenvalues. To have a detection of an adversarial example, it remains to determine whether the example is close in distance to a true (potentially unobserved) example: this is as hard as classifying the example correctly.
> > >
> > > In my opinion, **demonstrating the importance of the paper is crucial in this review process**. To me, the authors' findings are interesting, but the value of the findings is very unclear. I do appreciate the authors' effort in adding hundreds of more data points for the experiments, but the datasets contain thousands of data points, so we still cannot say that "the proposed theory reflects reality well" confidently. Another possible practical question is whether or not the findings will definitely lead to improvements in detection/defense methods. The authors claim that the findings will improve detection methods and refer to Tramer (2022), but there is no evidence that the findings will improve the detection methods, i.e., the detection methods are already exploiting the property unintentionally. If the authors can demonstrate this with experiments, it will be an important finding regardless of whether it can improve detection methods. If we can improve, then it is good, and just showing that the findings are already in existing detection methods is another important result that supports the importance of the findings.

---

> > > > ### Comment · Reviewer_EDzP · 2024-11-27
> > > >
> > > > **I'll maintain my rating of 5 (marginally below the acceptance threshold).** However, I honestly think the findings have enough potential to become a good paper, so I encourage the authors to keep pushing forward.

---

### Meta-Review · Area_Chair_u5HH · 2024-12-18

**Metareview:**

This submission received mixed reviews. While Reviewer azHL provided a positive rating, he/she did not actively champion the paper during the discussion phase. The remaining reviewers continued to express concerns, particularly regarding the paper's limited practical value. A stronger connection between the theoretical insights and practical relevance is necessary to enhance the impact of this work. Given the majority of reviewers' ratings and the current positioning of the paper, I regretfully recommend rejection.

**Additional Comments On Reviewer Discussion:**

While Reviewer azHL provided a positive rating, they did not actively champion the paper during the discussion phase. Reviewer 8xg  continued to express concerns, particularly about the paper's limited practical value during the discussion phase.

---

### Decision · Program_Chairs · 2025-01-22

Reject